# Shear-Wave-Elastography in Neurofibromatosis Type I

**DOI:** 10.3390/diagnostics12020360

**Published:** 2022-01-31

**Authors:** Deborah Staber, Julia Oppold, Alexander Grimm, Martin U. Schuhmann, Alyssa Romano, Justus Marquetand, Benedict Kleiser

**Affiliations:** 1Department of Epileptology, Hertie-Institute for Clinical Brain Research, University of Tübingen, 72076 Tubingen, Germany; deborah.staber@student.uni-tuebingen.de (D.S.); julia.oppold@student.uni-tuebingen.de (J.O.); alexander.grimm@med.uni-tuebingen.de (A.G.); alyssa.romano@student.uni-tuebingen.de (A.R.); benedict.kleiser@med.uni-tuebingen.de (B.K.); 2Section of Paediatric Neurosurgery, Department of Neurosurgery, University Hospital of Tübingen, Hoppe-Seyler-Str. 3, 72076 Tubingen, Germany; martin.schuhmann@med.uni-tuebingen.de; 3Department of Neurosurgery, University Hospital Tübingen, 72076 Tubingen, Germany; 4Centre for Neurofibromatosis at the Centre of Rare Diseases, University Hospital Tübingen, University of Tübingen, 72076 Tubingen, Germany; 5Department of Neural Dynamics and Magnetoencephalography, Hertie-Institute for Clinical Brain Research, University of Tübingen, 72076 Tubingen, Germany; 6MEG-Center, University of Tübingen, 72076 Tubingen, Germany

**Keywords:** SWE, neurofibromatosis, elastography, nerve, peripheral

## Abstract

Ultrasound shear wave elastography (SWE) is an increasingly used imaging modality that expands clinical ultrasound by measuring the elasticity of various tissues, such as the altered elasticity of tumors. Peripheral nerve tumors are rare, have been well-characterized by B-mode-ultrasound, but have not yet been investigated with SWE. Given the lack of studies, a first step would be to investigate homogeneous peripheral nerve tumors (PNTs), histologically neurofibromas or schwannomas, which can occur in multiple in neurofibromatosis type 1 and 2 (NF1 and 2), respectively. Hence, we measured shear wave velocity (SWV) in 30 PNTs of 11 patients with NF1 within the median nerve. The SWV in PNTs ranged between 2.8 ± 0.8 m/s and correlated with their width and approximate volume but not with their length or height. Furthermore, we determined the extent to which PNTs alter the SWV of the median nerve for three positions of the wrist joint: neutral (zero-degree), individual maximal flexion and maximal extension. Here, SWV was decreased in NF1 patients compared to age- and sex-matched controls (*p* = 0.029) during maximal wrist extension. We speculate that the presence of PNTs may have a biomechanical impact on peripheral nerves which has not been demonstrated yet.

## 1. Introduction

Elasticity of biological tissues change due to many factors, such as tumors, inflammation [1], and fibrosis [2]. Ultrasound shear wave elastography (SWE) provides measures of such altered elasticity and is already widely used for radiological staging of liver fibrosis [3], neoplastic changes in elasticity of breast cancer [4,5] and thyroid nodules [6]. Although SWE describes a different specific technique [7], the basic principle is generally as follows: Shear (or transversal) waves are induced acoustically by a transducer and propagate perpendicular to the main longitudinal transmission direction. As the speed of the propagating shear increases or decreases in relation to the elasticity of the tissue and can be tracked in real-time, quantitative two-dimensional maps of the shear wave velocity (SWV) or the shear modulus can be visualized (the shear modulus is quadratically related to the SWV). In neurology, especially in the field of peripheral nerves, SWE-studies are scarce; however, this non-invasive diagnostic has recently shown potential benefits in precisely diagnosing nerve compression syndromes such as carpal tunnel syndrome (CTS) [8] or cubital tunnel syndrome [9]. To further advance SWE in the area of peripheral nerves, it is reasonable to combine an already well-established disease entity for SWE, in this case tumors, with SWE of peripheral nerves. Due to the small scale of peripheral nerves, it is not surprising that nerve tumors are also typically small [10]. Therefore, we first investigated the feasibility of SWE and its potential utility for estimating the elasticity of peripheral nerve tumors (PNTs).

PNTs have been well-described in B-mode ultrasound [11,12]. For example, neurofibromas are typically characterized as a hypoechoic structure with possible central hyperechogenicity (“target sign”) [13]. A limitation of B-mode-ultrasound is that a clear differentiation between schwannoma, neurofibroma, or malignant degeneration remains, in some cases, difficult [13]. Winter et al. [10] proposed high-resolution B-mode-ultrasound to help the differentiation of benign and malignant peripheral nerve sheath tumors, but relied on PET and MRI imaging. Before exploring several types of PNTs with SWE, and considering the non-existence of studies in the field, a first step would be to investigate homogenous nerve tumors such as circumscribed non-plexiform neurofibromas and schwannomas in neurofibromatosis type 1 and 2. NF1 is an incurable autosomal dominant inherited disease with complete penetrance and wide variance in phenotypes, affecting mainly the skin and the nervous system [14]. The two most common forms are NF1 (peripheral neurofibromatosis, also known as Recklinghausen’s disease), followed by NF2 (central neurofibromatosis) [14].

Given the lack of studies in the field of PNTs, our study aims to expand the spectrum of SWE by investigating homogenous nerve tumors in NF1.

## 2. Materials and Methods

Right median nerves bearing previously known PNTs were investigated in 11 neurofibromatosis patients with NF1 using conventional B-Mode-ultrasound and SWE (Canon Aplio i800 device; 14 MHz linear transducer, i14LX5/PLI-1205BX, Canon Medical Systems, Neuss, Germany). For comparison, the right median nerve of 11 healthy age- and sex-matched healthy controls (HC) was also measured (for demographic details, refer to Table 1). HC were only included if they were over 18 years old and if they had no known neuromuscular disease. NF patients received diagnosis at an earlier date, typically clinically or radiographically. In only a few cases were the exact genetic data known. In addition, four patients with NF2 were also measured (see Appendix A), but their data were not further analyzed for the main analysis given the small number of cases.

Each patient or HC was seated on the same chair, with the right arm supported on a cushion in a standardized position with the back of the hand on the cushion and elbow flexed at 120°. First, the number of PNTs of the right median nerve was visually counted using B-mode ultrasound, starting from the flexor retinaculum of the hand on the right forearm. Then, the length, height, and width of the most visualizable PNTs were measured in the lower arm near to the wrist. Afterwards, the SWV of each of the best PNTs were measured lengthwise (Figure 1) with the following settings: size of the region of interest (ROI): 2, ROI shape: radius, frame rate: 1, time smoothing: 0 (no time averaging), map type: speed (display of the shear wave velocity in meters per second).

To assess if and to what extent PNTs alter the shear wave velocity (SWV) of the median nerve, all participants performed the following three positions in the wrist joint: neutral (zero-degree), individual maximal flexion and maximal extension (Figure 2). Additional measurements of the SWV of the median nerve were carried out in a PNT-free area of the median nerve and while maintaining the positioning of 3 cm proximal from the flexor retinaculum.

The statistical analysis of the collected data was performed with the Statistical Package for the Social Sciences (SPSS) version 27 for Windows, IBM Cooperation, Armonk, NY, USA. Our statistical design was as follows: First, we tested for normal distribution using the Shapiro–Wilk test. Normality could not be assumed in general, as the Shapiro–Wilk test was clearly significant for the measurement of the SWV in neurofibromatosis patients (W = 0.847, *p* = 0.039) and in healthy controls (W = 0.836, *p* = 0.028), so we continued using non-parametrical tests. Second, we tested whether there was a difference in SVV of the median nerve between the neurofibromatosis and healthy control groups, an unpaired sample, using the Mann–Whitney U test with a significance level of *p* < 0.05. We tested for each of the different positions (flexion, extension, and neutral) (3.2.). Third, we examined NF patients and HC separately using the Wilcoxon test for the paired sample. We examined whether flexion vs. extension, extension vs. neutral position, and neutral position vs. flexion made a significant difference (3.3.). Furthermore, the Spearman–Rho correlation coefficient was calculated between the number of PNTs and the SWV. Box plots were created using JMP 16.0.0 (SAS, Cary, NC, USA).

## 3. Results

### 3.1. SWE of PNTs in NF1

The measurement of SWE of PNTs proved feasible, showing an average SWV lengthwise of 2.8 m/s with a standard deviation of 0.8 m/s. In addition to the measured SWV, we also obtained the size of PNTs in the forearm’s median nerve. The mean size of the PNTs was 3.1 mm in height, 11.9 mm in length, and 4.9 mm in width. The approximate median volume was 44.69 mm^3^. There was a significant correlation between the measured SWV in PNTs and width (−0.697, *p* = 0.017) as well as between SWV in PNTs and approximate volume (−0.728, *p* = 0.026). No correlation was found in either the comparison between the measured SWV and length (−0.109, *p* = 0.75) or height (−0.528, *p* = 0.095). A correlation between the number of PNTs of the right median nerve of the forearm per patient and the SWV in extension was shown, with a correlation coefficient of −0.849 (*p* = 0.001) according to a Spearman–Rho correlation (Figure 1). However, two subjects had a value of approximately 100 tumors. In this case, the exact number was estimated since both patients had a large uncountable number of PNTs in the forearm’s median nerve. Excluding these two values, the analysis shows also significant correlation (r = −0.728, *p* = 0.026).

### 3.2. Comparison between Patients and Healthy Controls

The measured SWV in PNTs was compared to the SWV in the healthy median nerve to determine whether the tissue in PNTs results in a different SWV. PNTs showed a significant difference in SWV compared to the healthy median nerve in neutral position (U = 21.0, *p* = 0.008), as the mean SWV was lower in PNTs (2.8 m/s, SD 0.8 m/s) than in the HC’s median nerve (3.8 m/s, SD 1.1 m/s).

PNTs showed an impact on the median nerves’ SWV in the three positions of the wrist joint. The SWV was on average decreased in extension (U = 27.5, *p* = 0,029) and neutral position (U = 27.0, *p* = 0.027, refer to Table 2 and Figure 2). Both groups showed no significant difference in the flexion position (U = 47.0, *p* = 0.391).

### 3.3. Effect of Different Wrist Positions on the Median Nerve

In HC different wrist positions lead to significantly different SWV: SWV was higher in extension (5.8 m/s) than in flexion (3.2 m/s) as well as in neutral position (3.8 m/s) (extension vs. flexion (Z = −2.9, *p* < 0.001), extension vs. neutral (Z = −2.8, *p* < 0.002) and neutral vs. flexion (Z = −1.8, *p* = 0.071)). In NF patients, there was a similar trend, except for the difference between neutral position and flexion (extension vs. flexion (Z = −2.4, *p* = 0.014), extension vs. neutral (Z = −2.3, *p* = 0.019), neutral vs. flexion (Z = −0.5, *p* = 0.638)).

## 4. Discussion

This study on SWE of PNTs in neurofibromatosis takes a first step towards describing the elasticity of benign peripheral nerve tumors and their potential biomechanical influence on peripheral nerves. Due to the benign tumor entity and the homogenous ultrasound characteristics of neurofibromas and schwannomas [15], the disease NF1 can be regarded as a model to explore the potential of SWE for NF2 and PNTs. 

PNTs in this study exhibited an average SWV of 2.8 m/s. As there are no SWE-studies on peripheral nerve tumors, a comparison to non-nerve benign tumor seems reasonable. Azizi et al. [16] measured the SWV of 57 parathyroid adenomas and obtained a mean SWV of 2.02 m/s, compared to a healthy unaffected thyroid parenchyma SWV value of 2.77 m/s. Interestingly, the SWV was impartial to the adenoma size [16]. Likewise, the neurofibromas were also impartial to the NF approximate volume and length of tumor, but not width and height (see results). Like PNTs, adenomas are identified as predominately dense and having a solid body. However, unlike PNTs, adenomas are described by Kuo et al. [17] as without a clear boundary. PNT boundaries are rather unmistakable with ultrasound. Although similar in some respects and different in other respects, adenomas and PNTs could be used for comparison with SWV. This comparison would show that the PNTs have a faster SWV of just under 50% than the SWV of adenomas. This difference could be hypothetically due to certain characteristics, such as the hard-to-define boundaries of the adenomas, i.e., healthy tissue could be potentially included in the measurement, which would affect the SWV.

Liu et al. [18] also used SWV for the differentiation of tumors of the cervix uteri, with a mean of 3.53 m/s among 40 patients with benign tumors and 2.86 m/s in the HC, while the 138 patients with malignant tumors showed higher mean SWV at 4.91 m/s. Again, unlike the PNTs, the cervical carcinomas have unclear boundaries as well as an irregular shape. Regarding further studies differentiating benign and malignant tumor with ultrasound SWE, Ozturk et al. [19] and Zhong [20] investigated SWE in breast lesions. Benign breast lesions showed a mean SWV of 3.30 m/s, whereas malignant lesions had a mean SWV of 2.87 m/s [19]. Whether or not the mean calculated SWV lies on the lower end of the spectrum or the higher end is still questionable, as there are simply not enough large-scale SWE studies in benign homogenous tumors to facilitate comparison. That being said, amongst the above-mentioned studies, the NF mean SWV tends to run a little slower in comparison to other benign tumors. Further studies are needed to assess if ultrasound SWE can also be utilized to differentiate between benign and malignant tumors in peripheral nerves. Our results encourage such studies, as we show that the benign nerve tumor entities neurofibromas and schwannomas are in a similar range as other benign tumors. Consequently, it seems likely that malignant nerve tumors might also have—comparable to malignant non-nerve tumors—a higher SWV. In the light of sometimes conflicting studies, more studies are needed.

One can presume that the “stiffer” the nerve, the higher the estimated SWV. For example, in CTS patients showed an increased mean SWV in comparison to the HC [21]. In line with other SWE-studies about the median nerve, the SWV of the median nerve in neutral position in HC was 3.8 m/s; other studies report SWV between 3.1–4.0 m/s [22]. Interestingly, the SWV of the median nerve in NF patients showed a significantly lower SWV (2.8 m/s) in neutral position. This difference was even more evident when comparing different wrist joint positions. When the wrist joint is fully extended, the muscles around the median nerve are tensed while the nerve is also stretched due to the extension, resulting in higher compression of the nerve [22,23,24]. The opposite occurs when the wrist joint is fully flexed, relaxing the muscles around the nerve, creating less compression. The pathological nerve of the NF patients had a mean SWV of 2.8 m/s, meaning that the median nerves of the NF patients were less “stiff”. Therefore, a pathological nerve may lead to a differing bio-mechanical effect depending on the pathology. However, since SWE is still relatively young in the neurological field, more research with larger patient groups is strongly advised.

Further, influence of PNTs on the general stiffness of the nerve is possible. The stiffness of the median nerve differs depending on the position in the wrist, since the nerve is more under tension in extension than in flexion. We were able to confirm this in our measurements in healthy subjects. Thus, there was a higher stiffness in extension than in flexion or in neutral zero position.

In our study, patients with PNTs showed lower values than healthy controls when SWE of the median nerve was measured in the same position in extension and flexion. Thus, neurofibromas appear to have an effect on the stiffness of the nerve when the nerve is stretched.

One possible hypothesis is that PNTs have a similar effect as that of a pulley used in a block and tackle: when the nerve is stretched, the stretching force will be reduced by the neurofibroma, acting as a pulley. Consequently, the force due to the stretching will be reduced and hence also the “stiffness”, i.e., the SWV will be reduced. According to this hypothesis, the number of neurofibromas (or simply speaking, more pulleys) in the nerve should correlate with the SWV. In our small sample of 11 NF patients, we found a trend that the number of PNTs correlated negatively with the SWV, but we consider this correlation not representative due to the small sample size (see also Results). Here, studies with larger sample sizes are needed.

## 5. Conclusions

In conclusion, our study shows that the measurement of PNTs is feasible and shows a low SWV that is similar to other benign non-nerve tumors. Furthermore, we speculate that the presence of PNTs may have a biomechanical impact on peripheral nerves that has not yet been demonstrated.

## Figures and Tables

**Figure 1 diagnostics-12-00360-f001:**
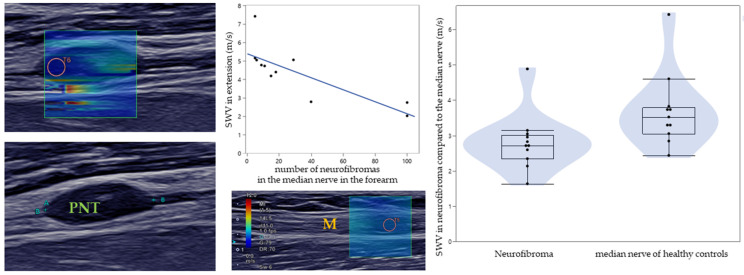
Measurement of a peripheral nerve tumor (PNT) in the median nerve. Left: Both B-mode images show the same PNT (above: with SWV, below: without SWV). Middle: Above, the diagram shows the distribution of SWV related to the number of PNTs. In blue, the correlation between the SWV in extension and the number of PNTs is shown. Below, a measurement of the SWV in the median nerve (M) is shown. Right: Boxplots comparing the SWV in neurofibroma with the SWV in the median nerve of healthy controls. The SWV is scaled from 0 m/s to 12 m/s by color (blue to red).

**Figure 2 diagnostics-12-00360-f002:**
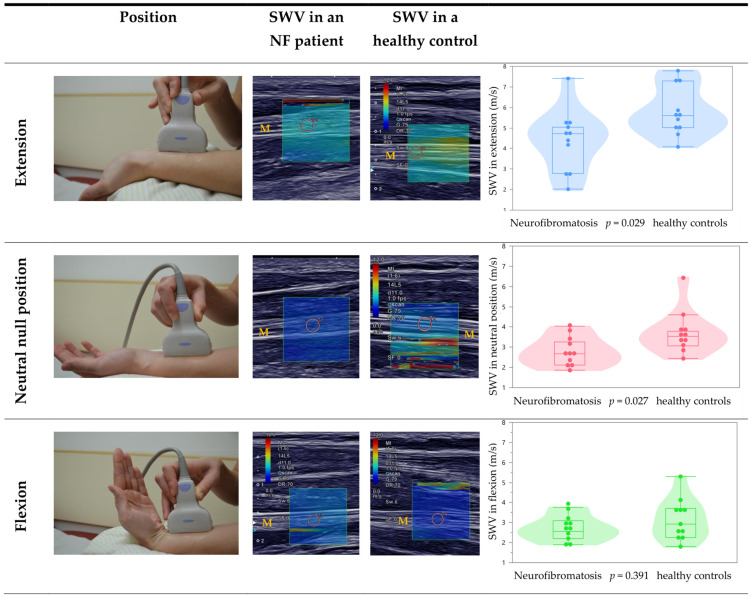
SWE of the median nerve in extension, neutral position, and flexion of the wrist in patients with neurofibromatosis type I and healthy controls. The shear wave velocity (SWV) is scaled from 0 m/s to 12 m/s by color (blue to red). In the middle, the position in the wrist is shown as an example. Note that the scaling is not shown in two images for the patient with neurofibromatosis (NF) due to export problems. The left column shows measurements in healthy controls, the right column is patients with neurofibromatosis type I. The orange “M” marks the median nerve in the ultrasound image. The orange circle marks the field of the measurement.

**Table 1 diagnostics-12-00360-t001:** Demographic details about patients with neurofibromatosis type I and healthy controls.

Parameter	Neurofibromatosis Type I	Healthy Controls
Age (years)	30.0 ± 10.6, 21/27/33 (21–51)	30.6 ± 11.0, 21/27/37 (20–53)
Body height (cm)	168.7 ± 11.3, 160/168/178 (153–187)	175.9 ± 8.0, 169/176/183 (163–189)
Body weight (kg)	69.6 ± 17.7, 57/64/78 (48–110)	74.6 ± 14.8, 62/75/85 (54–104)
Number of PNTs in the median nerve (forearm) *	30.7 ± 35.9, 6/15/40 (5–100) Median: 15	

Mean ± standard deviation, percentile 25%/50%/75% (range of all values). * including two patients with approximately 100 PNTs per nerve. PNTs: peripheral nerve tumors.

**Table 2 diagnostics-12-00360-t002:** Results for SWE in patients with neurofibromatosis type I and healthy controls.

Parameter	Healthy Controls (HC)	Neurofibromatosis Type I (NF I)	Comparision NF vs. HC
SWV of PNT/median nerve in lower arm (m/s)		2.8 ± 0.8, 2.4/2.7/3.0 (1.6–4.9)	U = 21.0, *p* = 0.008
SWV of median nerve in extension (m/s) ^+^	5.8 ± 1.2, 5.0/5.6/7.3 (4.1–7.8)	4.4 ± 1.5, 2.8/4.7/5.1 (2.0–7.4)	U = 27.5, *p* = 0.029
SWV of median nerve in neutral position (m/s) ^+^	3.8 ± 1.1, 3.1/3.5/3.8 (2.4–6.4)	2.8 ± 0.7, 2.1/2.7/3.3 (1.9–4.0)	U = 27.0, *p* = 0.027
SWV of median nerve in flexion (m/s) ^+^	3.2 ± 1.0, 2.3/2.9/3.7 (1.8–5.3)	2.7 ± 0.6, 2.2/2.6/3.1 (1.9–3.8)	U = 47.0, *p* = 0.391
Selected PNT length in lower arm (mm)		11.9 ± 8.0, 7.0/9.8/14.3 (1.4–27.8)	
Selected PNT height in lower arm (mm)	3.0 ± 0.4, 2.7/3.0/3.4 (2.5–3.7)	3.1 ± 2.1, 1.4/1.9/14.3 (1.0–7.4)	U = 54.5, *p* = 0.711
Selected width PNT in lower arm (mm)	4.4 ± 0.5, 4.1/4.3/4.7 (3.7–5.3)	4.5 ± 3.0, 2.4/3.0/6.0 (1.4–12.2)	U = 52.5, *p* = 0.617

Mean ± standard deviation, percentile 25%/50%/75% (range of all values). ^+^ different positions are demonstrated in Figure 2. SWV: shear wave velocity PNTs: peripheral nerve tumors.

## Data Availability

Raw data will be made available upon reasonable request.

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
