# Peer review of "Shear-Wave-Elastography in Neurofibromatosis Type I"

_diagnostics, 2022, doi:10.3390/diagnostics12020360_

Round 1

Reviewer 1 Report

Review of the publication “Shear-Wave-Elastography in Neurofibromatosis Associated Peripheral Nerve Tumors” by Staber and co-workers

Summary:

In their study, the authors want to characterise the shear wave elastography parameter (SWE) of peripheral nerve tumors. As a model disease, they selected patients suffering from NF Type 1 and 2 and compared them to age and gender matched controls. They searched for peripheral nerve tumors along the median nerve. The SWE was measured along the nerve and at the location of tumors.

General:

SWE is a promising technique to extend the diagnostic gain of nerve ultrasound imaging. Therefore  the authors study addresses an important research aspect. However, the study and manuscript needs major revision before it is suitable for publication.

First I will address my major general concerns and then I will go thru the manuscript point by point:

Study group:

The mechanisms how tumors are developing in NF Type I patients is hardly understood.  Most likely an alteration in the RAS pathway (Omim: 162200 ) due to a mutation in the neurofibromin gene is involved.

The situation in NF Type II is much more clear. Here a critical regulator in cell proliferation – merlin -is altered (Omim: 101000).

The proposed study wants to analyse the SWE feature of nerve tumors. In order to unique disease model I would concentrated on one of the diseases. Given the larger number of subject with NF Type I I, I would focus on NF type I.

Measurement paradigm

The measurement paradigm sound reasonable to me. Testing the nerve in different hand positions ( hand flexed, neutral and extended ) allows to see how shear force inside the nerve alter the velocity parameter.

Statistics

This is by far the weakest point of the study.

“Has this project been discussed with an expert in the field of medical statistics? “

First, given the small numbers, I would suggest to use median and quantile instead median and STDEV throw out the manuscript.

 The study data was correctly tested for normality but then a series of non-parametric test follow ( Wilcoxon and Mann – Whiteny Test) without a clear statistical aim.

I would suggest to first clearly layout the statistical test concept in the method section and then apply the test in the result section one after the other.

Results

The authors do have important data to present and share, but unfortunately the important scatter / box plots are missing. Especially the data of the SWE speed in tumors should be presented in a much more clearer and strict manner.

This data would help the clinical in the field to know what to look for in patients with NF1 and might help to detect tumors early.

Discussion

I am happy to review the discussion in great detail once the above points have been addressed.

Point by point:

  • Check sentence Line 61 / 62
  • Table 1 add Genetic data
  • Was one PNT selected in the lower and one in the upper arm? Or were the two best PNTs which might be both located in the lower arm?
  • Figures -> check numbering
  • Results: Add box plots of the results comparing NF1 and normal data
  • Table 2: Don’t mix parametric and nonparametric measures and tests

Author Response

We would like to thank the reviewer for addressing these important points, which helped to improve the overall quality of our manuscript.

Summary:

In their study, the authors want to characterise the shear wave elastography parameter (SWE) of peripheral nerve tumors. As a model disease, they selected patients suffering from NF Type 1 and 2 and compared them to age and gender matched controls. They searched for peripheral nerve tumors along the median nerve. The SWE was measured along the nerve and at the location of tumors.

General:

SWE is a promising technique to extend the diagnostic gain of nerve ultrasound imaging. Therefore  the authors study addresses an important research aspect. However, the study and manuscript needs major revision before it is suitable for publication.

First I will address my major general concerns and then I will go thru the manuscript point by point:

Study group:

The mechanisms how tumors are developing in NF Type I patients is hardly understood.  Most likely an alteration in the RAS pathway (Omim: 162200 ) due to a mutation in the neurofibromin gene is involved.

The situation in NF Type II is much more clear. Here a critical regulator in cell proliferation – merlin -is altered (Omim: 101000).

The proposed study wants to analyse the SWE feature of nerve tumors. In order to unique disease model I would concentrated on one of the diseases. Given the larger number of subject with NF Type I I, I would focus on NF type I.

  • Answer: We agree that the pathomechanism is different between neurofibromatosis type I and type II. Therefore, in our new version of this paper, we performed the statistics only with patient with neurofibromatosis type I. As a next step, a detailed investigation of different tumors or space-occupying lesions would be useful - in particular, differences between patients with NF type I and NF type II would be of interest. Our supplemental Table 1 distinguishes between patients with NF type I and NF type II. There were no significant differences, although it should be noted that measurement results were available in only 4 patients with NF type II.
  • In the course of reanalyzing the study (NF1 only), we updated all figures and tables in the manuscript. Of note, this new and extensive analysis did not change the overall results as before, but we thank the reviewer for making the manuscript and analysis more clear. In our follow-up-study we will take care of the difference between NF1 and NF2 and try to study a larger n of patients. We hope that our efforts are suffice.

Measurement paradigm

The measurement paradigm sound reasonable to me. Testing the nerve in different hand positions ( hand flexed, neutral and extended ) allows to see how shear force inside the nerve alter the velocity parameter.

Statistics

This is by far the weakest point of the study.

“Has this project been discussed with an expert in the field of medical statistics? “

First, given the small numbers, I would suggest to use median and quantile instead median and STDEV throw out the manuscript.

  • Answer: This project has been discussed with an expert in the field of medical statistics and we agree with the reviewer that the representation of a non-normally distributed group is better done using quantiles. In our new version we also show the quantiles.

The study data was correctly tested for normality but then a series of non-parametric test follow ( Wilcoxon and Mann – Whiteny Test) without a clear statistical aim.

  • Answer: The test for a normal distribution could not confirm a normal distribution. We therefore decided to test by means of non-parametric tests. On the one hand, the question arises whether significant differences exist between the healthy subjects and the diseased patients. Therefore, we are dealing with unpaired samples, so that we decided to use the Mann-Whitney U-test for such questions. Another question was whether there were significant differences between different measurement time points (flexion, neutral position, or extension) within a group (HC or diseased patients). Therefore, we decided to use the Wilcoxon test.

I would suggest to first clearly layout the statistical test concept in the method section and then apply the test in the result section one after the other.

  • Answer: We have revised the methods section and included an instructive step-by-step statistics concept to contribute to the field:
  1. Testing normal distribution (Shapiro-Wilk). Normality could not be assumed, so we continued using non-parametrical tests.
  2. Comparing the SWV of the unpaired sample NF and HC using the Mann-Whitney U-Test in different positions (flexion, extension, and neutral) (3.2.).
  3. Check within the group (paired sample), Wilcoxon Test, whether flexion vs. extension, extension vs. neutral position, and neutral position vs. flexion made a significant difference. Further, we examined the two groups for overall significant differences between flexion, extension, and neutral position respectively for NF and HC using the Friedman test. (3.3.)

 Now we write:

“The statistical analysis of the collected data was performed with the Statistical Package for the Social Sciences (SPSS) version 27 for Windows. Our statistical design was as follows: First, we tested for normal distribution using the Shapiro-Wilk test. Normality could not be assumed in general, as the Shapiro-Wilk test was clearly sig-nificant for the measurement of the SWV in neurofibromatosis patients (W=0.847, p=0.039) and in healthy controls (W=0.836, p=0.028) so we continued using non-parametrical tests. Second, we tested whether there was a difference in SVV of the median nerve between the neurofibromatosis and healthy control groups, an unpaired sample, using the Mann-Whitney U test with a significance level of p< 0.05. We tested in the different positions (flexion, extension, and neutral) (3.2.). Third, we examined NF patients and HC separately using the Wilcoxon test for the paired sample. We ex-amined whether flexion vs. extension, extension vs. neutral position, and neutral posi-tion vs. flexion made a significant difference (3.3.). Furthermore, the Spearman-Rho correlation coefficient was calculated between the number of PNTs and the SWV. Box plots were created using JMP 16.0.0 (SAS, Cary, NC, USA).”

Results

The authors do have important data to present and share, but unfortunately the important scatter / box plots are missing. Especially the data of the SWE speed in tumors should be presented in a much more clearer and strict manner. This data would help the clinical in the field to know what to look for in patients with NF1 and might help to detect tumors early.

  • Answer: For the comparison between different positions we created boxplots (see Figure 1). We also added a figure showing boxplots of the SWV of the tumors themselves (see Figure 2).

 Now Figure 2 is updated:

Figure 2. Measurement of a PNT in the median nerve. Left: both B- mode images show the same PNT, above with SWV, below without the SWV. Middle: Above, the diagram shows the distribution of SWV related to the number of PNTs. In blue, the correlation between the SWV in extension and the number of PNTs is shown. Below, a measurement of the SWV in the median nerve (M) is shown. Right: Boxplots, comparing the SWV in neurofibroma with the SWV in the median nerve of healthy controls. The SWV is scaled from 0m/s to 12m/s by color (blue to red).

Discussion

I am happy to review the discussion in great detail once the above points have been addressed.

Point by point:

Check sentence Line 61 / 62

  • Answer: We changed the sentence to: “NF1 are incurable autosomal dominant inherited diseases with complete penetrance with a wide variance of phenotypes, affecting mainly the skin and the nervous system [14]”.

Table 1 add Genetic data

  • Answer: We thank the reviewer for bringing up this matter. Unfortunately, the genetic data was not available for all patients since the diagnosis of NF1 or NF2 was already made clinically/radiographically. Therefore, we did not add an additional table of genetic data. Of note, only in 4 of 15 the patient group (NF1&2) were available. To address the matter of genetic data, we will consider it in a follow-up study and now write in “NF patients received diagnosis at an earlier date typically clinically or radiographically. In only a few cases the exact genetic data was known.”

Was one PNT selected in the lower and one in the upper arm? Or were the two best PNTs which might be both located in the lower arm?

  • Answer: Unfortunately, an error crept into the method section. For the data presented here, PNTs were only measured on the forearm near the wrist. We apologize for this confusion and corrected this in the paper.
  • Now we write: “Firstly, the amount of PNTs of the right median nerve was visually counted using B-mode ultrasound, starting from the flexor retinaculum of the hand for the right forearm. Then, the length, height and width of the best visualizable PNTs were measured in the lower arm near to the wrist. Afterwards, the SWV of each of the best PNTs were measured lengthwise (Figure 2) with the following settings…”

Figures -> check numbering

  • Answer: Checked the numbers. Now the references to the figures should be fine.

Results: Add box plots of the results comparing NF1 and normal data

  • Answer: Figure 2 is now supplemented with the boxplot showing the neurofibromas of NF1 and the healthy controls.

Table 2: Don’t mix parametric and nonparametric measures and tests

  • Answer: In the patient group presented here, there are some patients who have very many neurofibromas. The exact counting of neurofibromas is therefore very prone to error, so that the exact number of neurofibromas is difficult to determine and a total of 100 neurofibromas was assumed to be the most likely.

Reviewer 2 Report

The authors present a nice paper on the use of SWE to evaluate peripheral nerve tumors in a set population for the first time.

I think the authors might still extend their discussion on how this technique may be of use in the future for both NF populations as well as sporadic nerve tumors. Is this technique also useful in tumors that are more deeply seated (but still visible on ultrasound)? How would depth influence the results? Do you think there are differences in values between NF and sporadic patients? Are there differences between schwannoma and neurofibromas?

Author Response

Reviewer 2:

The authors present a nice paper on the use of SWE to evaluate peripheral nerve tumors in a set population for the first time.

I think the authors might still extend their discussion on how this technique may be of use in the future for both NF populations as well as sporadic nerve tumors. Is this technique also useful in tumors that are more deeply seated (but still visible on ultrasound)? How would depth influence the results?

  • Answer: We thank the reviewer for the notice. In fact, it is known that the measured value SWV depends on both the depth of the measuring point in the tissue and the frequency of the probe (see for example Chang et al., 2012). An effect is therefore quite conceivable. This should be investigated in further studies. In the present work, we tried to give a first impression for SWV and neural tumors. We deliberately measured close to the wrist. We think that this way, if possible, values could be measured in different subjects and patients always under the same conditions. Last but not least, the median nerve already runs rather superficially at this measurement point, so that the depth of the measurement point did not vary too much depending on other structures such as muscles.

Do you think there are differences in values between NF and sporadic patients? Are there differences between schwannoma and neurofibromas?

  • Answer: This is a question which should definitely be investigated in further work. As already mentioned in the discussion, a distinction between benign and malignant tumors is conceivable. In this regard, SWV as an additional technique could refine ultrasound diagnostics and improve sensitivity as well as specificity to determine the exact neural tumor. At present, however, such comparisons are not available, so that this is currently pure conjecture, which should definitely be investigated in further studies.

Chang S, Kim MJ, Kim J, Lee MJ. Variability of shear wave velocity using different frequencies in acoustic radiation force impulse (ARFI) elastography: a phantom and normal liver study. Ultraschall Med. 2013 Jun;34(3):260-5. doi: 10.1055/s-0032-1313008. Epub 2012 Sep 21. PMID: 23023455.
